# Generation of the Human Pluripotent Stem-Cell-Derived Astrocyte Model with Forebrain Identity

**DOI:** 10.3390/brainsci11020209

**Published:** 2021-02-09

**Authors:** Ulla-Kaisa Peteri, Juho Pitkonen, Kagistia Hana Utami, Jere Paavola, Laurent Roybon, Mahmoud A. Pouladi, Maija L. Castrén

**Affiliations:** 1Faculty of Medicine, Department of Physiology, University of Helsinki, P.O. Box 63, FI-00290 Helsinki, Finland; ulla-kaisa.peteri@helsinki.fi (U.-K.P.); Juho.Pitkonen@helsinki.fi (J.P.); 2Translational Laboratory in Genetic Medicine, Agency for Science, Technology and Research, Singapore (A*STAR), 8A Biomedical Grove, Immunos, Level 5, Singapore 138648, Singapore; kagistia@thetlgm.com (K.H.U.); pouladi@thetlgm.com (M.A.P.); 3Minerva Foundation Institute for Medical Research, Biomedicum 2U, Tukholmankatu 8, FI-00290 Helsinki, Finland; jere.paavola@helsinki.fi; 4Stem Cell Laboratory for CNS Disease Modeling, Department of Experimental Medical Science, Lund University, SE-221 84 Lund, Sweden; laurent.roybon@med.lu.se; 5MultiPark and the Lund Stem Cell Center, Lund University, SE-221 84 Lund, Sweden

**Keywords:** pluripotent stem cells, differentiation, astrocytes, patterning, fragile X syndrome

## Abstract

Astrocytes form functionally and morphologically distinct populations of cells with brain-region-specific properties. Human pluripotent stem cells (hPSCs) offer possibilities to generate astroglia for studies investigating mechanisms governing the emergence of astrocytic diversity. We established a method to generate human astrocytes from hPSCs with forebrain patterning and final specification with ciliary neurotrophic factor (CNTF). Transcriptome profiling and gene enrichment analysis monitored the sequential expression of genes determining astrocyte differentiation and confirmed activation of forebrain differentiation pathways at Day 30 (D30) and D60 of differentiation in vitro. More than 90% of astrocytes aged D95 in vitro co-expressed the astrocytic markers glial fibrillary acidic protein (GFAP) and S100β. Intracellular calcium responses to ATP indicated differentiation of the functional astrocyte population with constitutive monocyte chemoattractant protein-1 (MCP-1/CCL2) and tissue inhibitor of metalloproteinases-2 (TIMP-2) expression. The method was reproducible across several hPSC lines, and the data demonstrated the usefulness of forebrain astrocyte modeling in research investigating forebrain pathology.

## 1. Introduction

Astrocytes represent a major cell population in the mammalian brain [1] and are responsible for a wide range of functions, including the maintenance of extracellular homeostasis, co-ordination of the blood–brain barrier (BBB) permeability and an essential contribution to regulation of synaptic function [2,3,4]. Gliogenesis is induced by factors that are tightly linked to the control of neurogenesis, and non-neuronal cells are largely generated after birth [5,6,7,8]. Cues mediating the process vary between brain regions, and in the forebrain, one of the key mechanisms mediating the generation of glia and forebrain composition is Notch1 signaling [9]. Cortical precursors vary in their responses to Notch1 signaling, and its activity inhibits the generation of neurons and later, during gliogenesis, promotes the generation of glial fibrillary acidic protein (GFAP)-expressing cells [10].

Like neurons, astrocytes are known to consist of functionally and morphologically distinct populations of cells with region-specific properties [11,12,13,14]. Astrocytes obtain their morphological and molecular region-specific properties upon maturation and neurons influence the maturation process [15,16,17,18]. Furthermore, glia exhibit specific properties and some of the astrocyte populations found in the human brain are not represented in rodents [19]. The study conducted by Han and colleagues demonstrated functional differences in human and rodent astrocytes by showing improved cognitive abilities in mice with engrafted human astrocytes [20]. When studying astrocyte function, it is important to address variability in the cell population within distinct regions as well as between species.

Astrocyte abnormalities are increasingly linked to several CNS disorders [8,21,22]. This calls for a demand to develop novel tools to model the role of astrocytes in disease pathologies. Human pluripotent stem cells (hPSCs) have provided invaluable means to generate human astrocytes for disease modelling [23]. Accordingly, an increasing number of astrocyte differentiation methods have emerged aiming at standardized methodology [24,25,26,27,28,29]. However, due to the diversity of astrocytes, new models need to be developed and tested.

Here, we describe a method that generates human astrocytes with forebrain identity. Transcriptomic analysis performed at selected stages of the differentiation confirmed that the forebrain patterning was persistent. Generated astrocytes were functional and positive for astrocytic markers, suggesting that the astrocyte model could be useful for studies of disease-specific abnormalities of astrocytes.

## 2. Materials and Methods

### 2.1. Generation of hPSC-Derived Astrocytes

Human astrocytes were generated from human embryonic stem cell line H1 and hiPSC lines HEL23.3, HEL24.3, PO2/UEF-3A, and PO4/UEF-3B derived from healthy donors [30,31,32,33]. The research using hPSCs has been approved by the Ethical Committee of the Hospital District of Helsinki and Uusimaa. The hPSCs were maintained on Matrigel-coated plates in Essential 8 (E8) medium (Thermo Fisher Scientific, Waltham, MA, USA). For astrocyte differentiation, 80% confluent pluripotent stem cells were dissociated with 0.5 mM ethylenediaminetetraacetic acid (EDTA) (Invitrogen, Carlsbad, CA, USA) and plated at a 1:1 ratio on a low-attachment 6-well plate in E8 with 20 ng/mL human bFGF2 (Peprotech, Somerset County, NJ, USA) and 20 µM Rho kinase inhibitor (Sigma-Aldrich, St. Louis, MO, USA). On the following day, medium was changed to Neuronal Induction Medium (NIM) (Advanced DMEM/F12, 1X N2, 2 mM L-glutamine, 1X Non-essential Amino Acid (NEAA) and 1X P/S (all from Thermo Fisher Scientific)) supplemented with 0.1 µM LDN-193189 (Stemgent, Cambridge, MA, USA), 1 µM cyclopamine (Sigma-Aldrich), 10 µM SB-431542 (Sigma-Aldrich) and 0.5 µg/mL DKK1 (Peprotech). The patterning and neural induction took place between D0 and D12 with medium changes every second day.

Neural progenitor cells (NPCs) multiplied and matured between D12 and D60. Until D30, NPCs were cultured in NIM supplemented with 20 ng/mL brain-derived neurotrophic factor (BDNF; Peprotech) with medium changes every second day. Thereafter, culture medium was changed from NIM to neurosphere medium (NS) (Advanced DMEM/F12, 1X B27 -RA, 1X L-glutamate, 1X NEAA, 1X P/S (all from Thermo Fisher Scientific)) supplemented with 20 ng/mL basic fibroblast growth factor (bFGF2) and 20 ng/mL epidermal growth factor (EGF) (both from Peprotech) with medium changes twice a week and with growth factors added three times a week. At this point, the spheres were dissociated approximately once per week manually. At D60, the spheres were dissociated and progenitors plated on poly-ornithine/laminin-coated (Sigma-Aldrich) culture plates at 1:1 ratio in NS medium supplemented with 20 ng/mL ciliary neurotrophic factor (CNTF) (Peprotech). Medium was changed twice per week and the cells passaged with Trypsin-EDTA (0.05%) (Thermo Fisher Scientific) approximately once per week and seeded at 20,000 cells/cm^2^. After D75, once the cells had acquired astrocyte morphology they were maintained on Matrigel-coated culture plates. Astrocyte morphology changed upon maturation, and all functional experiments were performed after D95.

### 2.2. Immunocytochemistry

For the immunostaining of embryoid bodies (EBs), cells were fixed with 4% paraformaldehyde (PFA) for 45 min and washed three times with phosphate-buffered saline (PBS, pH 7.4). Fixed EBs were incubated in PBS overnight at 4 °C. On the following day, PBS was replaced with 30% sucrose in PBS and cell clusters were incubated at 4 °C until they had settled to the bottom of the tubes. Sucrose solution was replaced by OCT mounting medium (VWR) and embedded EBs were snap-freezed. EBs were cryosectioned (Leica cryostat) into 20 µm sections, which were placed directly on Superfrost Plus object glasses (Thermo Fisher). EB sections were blocked with 10% normal donkey serum (NDS)-0.05% Triton X-100 in PBS for 1 h. Sections were incubated overnight at +4 °C with primary antibodies rabbit anti-PAX6 (BioLegend, San Diego, CA, USA, PRB-278P, 1:500) and chicken anti-TBR2 (Millipore, Burlington, CA, USA, AB15894, 1:500) and 1 h at RT with secondary antibodies (Alexa Fluor 555 donkey anti-rabbit (Invitrogen, Carlsbad, CA, USA, A31572, 1:800) and Alexa Fluor 647 donkey anti-chicken (Millipore, AP1495A6, 1:500). Nuclei were stained with 4´6-diamino-2-phenylindole (DAPI; Sigma, D9542, 1:10,000) for 5 min.

For the astrocyte marker analysis, cells were plated on ViewPlate-96 (PerkinElmer, Waltham, MA, USA) with 5000 cells per well. Two days later, cells were fixed with 4% PFA for 15 min at RT and washed three times with PBS. Prior to incubation with primary antibody overnight at +4 °C and with secondary antibody for 1 h at RT, cells were blocked and permeabilized with 10% normal goat serum (NGS)-0.05% Triton X-100 in PBS for 1 h. Primary antibodies used were chicken anti-GFAP (Abcam ab4674, 1:1000), rabbit anti-SOX9 (Cell Signaling, 82,630, 1:400), rabbit anti NF1A (Active Motif, 39,397, 1:500), mouse anti-S100β (Sigma-Aldrich, S-2532, 1:500) and mouse anti-MAP2 (Sigma-Aldrich, M4403, 1:2500). Secondary antibodies used were Alexa Fluor 488 goat anti-chicken, Alexa Fluor 546 goat anti-rabbit and Alexa Fluor 635 goat anti-mouse (1:500, Invitrogen). After final washes, nuclei were counterstained with DAPI (Sigma-Aldrich). The cells were imaged with Thermo Scientific CellInsight instrument and analyzed using ImageJ and CellProfiler software.

### 2.3. RNA Sequencing and Analysis

RNA-seq data were generated using triplicate samples of hPSC (H1)-derived cells at defined time points during astrocyte differentiation and NextSeq500 platform. The sequencing reads were filtered based on quality and length, and, after adapter removal, the reads were aligned to the human genome using Star Aligner (version 2.5.0b). Following read alignment, HTSeq read counting and DESeq differential expression analysis was performed relative to D0 samples.

### 2.4. RNA Expression Analysis with Quantitative Real Time PCR (qRT-PCR)

Total RNA was extracted using a NucleoSpin RNA kit (Qiagen, Hilden, Germany), and mRNA reverse transcribed into complementary cDNA using a iScriptTM cDNA synthesis kit (Bio-Rad, Hercules, CA, USA, #170-8891). Using cDNA as a template, RT-qPCR was performed with HOT FIREPol^®^ EvaGreen^®^ qPCR Mix Plus (Solis BioDyne, Tartu, Estonia) and LightCycler^®^ 480 (Roche, Basel, Switzerland) for 45 cycles of 95 °C for 15 s, 62 °C for 20 s and 72 °C for 20 s. Gene expressions were analyzed with the ΔΔCt method using GAPDH expression and values from undifferentiated pluripotent cells for normalization. Primer sequences are listed in Appendix A.

### 2.5. Calcium Imaging

For calcium imaging, hPSC-derived astrocytes were plated on coverslips in a 12-well plate with 50,000 cells per well and imaged on the following day. Prior to imaging, the cells were incubated with 4 µM Fura-2 (HelloBio, Bristol, UK, HB0780) at 37 °C for 30 min. The coverslip was then adjusted to SA-OLY/20 adapter and placed in the recording chamber with continuous perfusion with 37 °C HEPES-buffered medium (HBM) (137 mM NaCl, 5 mM KCl, 0.44 KH_2_PO_4_, 4.2 mM NaHCO_3_, 1 mM MgCl_2_, 1 mM CaCl_2_ and 10 mM glucose, pH 7.4). To elicit ATP responses, the cells were exposed to 100 µM ATP in HBM for 30 s. Calcium measurements were performed using an Inverted IX70 microscope (Olympus Corporation, Hamburg, Germany). Astrocytes were visualized with (UApo/340) ×10 air objective (Olympus) and images acquired with Hamamatsu ORCA-Flash 4.0 camera. The cells were excited with 340 and 380 nm wavelengths and the emission recorded at 505 nm. Ca^2+^ transients are represented as Fura-2 340/380 ratio units. Data analysis was performed using HCImage and Matlab.

### 2.6. Cytokine Array

To screen for the secretion of cytokines in astrocytes, astrocyte-conditioned medium (ACM) was analyzed using Human Cytokine Antibody Array (Abcam, Cambridge, UK, Ab133998). ACM was prepared by plating the cells on Matrigel-coated T25 flasks at a density of 20,000 cells per cm^2^ and grown for 7 days. On the day prior to collecting the samples, medium was replaced with 5 mL of fresh NS. Collected medium was filtered through a 0.22 µm filter and stored at −80 °C. For the analysis, media from four iPSC-derived astrocyte lines were pooled and array performed with undiluted medium according to the manufacturer’s instructions. Chemiluminescent detection was performed with a ChemiDoc imaging system (Bio-Rad Finland, Helsinki, Finland) and densitometric data obtained using ImageJ software. Mean intensities of negative and positive control spots were used for background correction and normalization, respectively.

## 3. Results

The generation of astrocytes using the differentiation protocol introduced herein was divided in three stages (Figure 1a). First, the neuronal differentiation of hPSCs was induced using dual inhibition of SMAD signaling in the presence of SB431542 and LDN193189 [34]. The patterning towards forebrain identity was facilitated by cyclopamine and DKK1, inhibitors of sonic hedgehog (SHH) and WNT signaling [35], respectively. The generated neural/astroglial progenitors were allowed to expand between Day 12 (D12) and D60 first in the presence of BDNF and thereafter with EGF and bFGF. At the third phase, progenitors were dissociated and plated for astrocyte specification and elimination of neurons with CNTF [36] and regular passaging of the cells, respectively. In addition to morphological changes of astrocytes upon maturation, the number of progenitors seen as small bright cells with a halo decreased (Figure 1b). The differentiation was monitored by the transcriptome profiling of undifferentiated hPSCs (D-1) and at D0, D12, D30, D60, and D95 of differentiation. The gene expression pattern of D95 astrocytes was clearly separated from that of undifferentiated progenitors and from the clusters of cells at D12, D30, and D60 of differentiation in the principal component analysis (PCA) (Figure 1c). Comparison of the transcriptome profile of astrocytes aged D95 with published databases verified the differentiation of hPSCs into human astrocyte lineage cells by showing that the transcriptome profile of D95 astrocytes was similar to that of human astrocytes isolated from the embryonic brain [37] as well as to that of hPSC-derived astrocytes published previously [27] (Figure 1c).

### 3.1. Patterning and Differentiation of Neural Progenitors

Successful neuralization was seen as the formation of neural rosette structures [38] within spheres (Figure 2a). Immunocytochemical and qPCR analysis of the spheres established the expression of neuronal progenitor markers PAX6 and TBR2 in primary and secondary progenitors that were organized radially in a way resembling the dorsal ventral axis of columnar neuroepithelial cells (Figure 2a). Transcriptome profiling confirmed differentiation towards forebrain identity. Gene ontology (GO) analysis [39] revealed that genes involved in forebrain development were enriched at D12 (Figure 2b). EMX1 has an important role in the regulation of dorsal telencephalic development [40], and its expression was increased from the beginning of the progenitor differentiation. Expression of forebrain markers Forkhead Box G1 (FOXG1), LIM homeobox 2 (LHX2), and DLX1 [41] was induced from D12 onwards (Figure 2c) and remained relatively strong until the end of the differentiation, while the expression of midbrain and hindbrain markers was low throughout the differentiation. The GO analysis at D60 demonstrated that forebrain development was still the most persistent biological process (Figure 2d), supporting the notion that the initial regional specification was successful and had long-lasting effects on the gene expression in the differentiating cells.

The Notch signaling, together with pro-glial cell transcription factors, orchestrates differentiation and the fate determination of neural progenitors to astrocytes [9]. Notch signaling regulates the responsiveness to SHH signaling that controls SOX9 expression, which is critical for the switch from neurogenesis to gliogenesis [42]. Expression of SOX9 was significantly increased from D30 onwards in hPSC-derived cells (Figure 2e). Also, expression of NF1A that regulates the onset of gliogenesis [37] was identified from D30 onward (Figure 2e). In the expression profiling, altogether, 4233 genes showed differential expression at D30 when compared to the gene expression pattern at D0. The set of genes included early astrocyte markers SLIT-1, FGF3 and GLI3, whose expression peaked at D30 (Figure 2e). Nestin remained without greater variation in the expression levels during astrocyte differentiation (Figure 2e).

### 3.2. Astrocyte Specification

At D60, the spheres were dissociated and plated into adherent culture. The final specification of astrocytes was performed in the presence of CNTF that is a potent inducer of astrocyte maturation [36]. We differentiated the embryonic stem cell (ESC)-derived H1 hPSC line and three control human induced pluripotent stem cell (hiPSC) lines to astrocytes and found that, by D75, cells were positive for the selected astrocyte markers (SOX9, NFA1, S100β, and GFAP) (Figure 3a,b). Although identical proportions of cells were positive for these astrocyte markers at D75 and D95, maturation of astrocytes could be seen as a change in cellular morphology. At Day 75, the cells had smaller cell bodies and longer processes, whereas, upon differentiation, they turned larger and more cuboidal (Figure 3a). Further quantification at D75 and D95 revealed that more than 90% of total cells were positive for GFAP (Figure 3b). Most GFAP^+^ cells were also immunopositive for S100β (94.94% ± 5.43 % of 15,448 cells at D75 and 97.78% ± 3.24% of 7449 cells at D95) (Figure 3c).

### 3.3. Properties of hPSC-Derived Astrocytes

Astrocytes aged D95 expressed several astrocyte markers [43] (Figure 4a,b). Expression of established astrocyte markers Aquaporin 4 (AQP4), and Vimentin (VIM), as well as expression of genes involved in synapse formation, maturation, and maintenance through astrocytic secretion of SPARC-like protein 1 (SPARCL1) and Neuronal Cell Adhesion Molecule (NRCAM) were enriched in RNA-seq data. The molecular signature of D95 astrocytes also included high expression of a number of genes that were previously shown expressed in astrocytes derived from human neural progenitors isolated from the fetal brain CD44, CDKN2B, LGALS3, LHX2, LMO2, PPRX1, and TGFB3 [37] (Figure 4a).

Strong expression of GFAP in D95 astrocytes was seen in the transcriptome analysis (Figure 4a) and was confirmed by RT-qPCR (Figure 4b). The pattern of GFAP expression relative to expression levels of S100β and excitatory amino acid transporter EAAT1, was similar in astrocytes differentiated from different hPSC lines (Figure 4b), whereas mRNA for the glutamate transporter EAAT2 was not detectable. Given that neurons are shown to be crucial for the expression of EAAT2/GLT-1 in astrocytes [44,45], a low level of EAAT2 expression supported the notion that the culture was largely free of neurons.

In transcriptome analysis, altogether, 1500 genes were found to be differentially expressed when the mRNA expression profile of D95 astrocytes was compared to that of cells at D0. Most of the regulated genes showed decreased expression. The expression of 240 genes was increased and these genes were enriched in ontology clusters of sensory organ development, pattern specification process, extracellular matrix organization, and interferon signaling (Figure 4c). The forebrain development gene cluster included 17 genes and gliogenesis included 23 genes, suggesting that the gene expression pattern of the mature astrocytes reflected changes instructed by the early forebrain patterning.

Calcium imaging analysis demonstrated functional responses of D95 astrocytes. Intracellular calcium response was seen in 86% of cells after exposure to 100 µM ATP (Figure 5a,b). Average response was relatively slow and long lasting compared to the typical response elicited in neurons throughout the cell population (Figure 5c), which further indicated that, after D95, the cultures were largely composed of astrocytes. In conclusion, the method generated highly pure cultures of functional astrocytes, providing an ideal tool for disease modeling, to study the innate properties of astrocytes as well as neuron-astroglia interactions in co-cultures.

One of the key functions of astrocytes in the central nervous system (CNS) is the modulation of immune response. Several genes in the cytokine-mediated signaling pathway (20/798) were expressed in human astrocyte cultures aged D95. To identify components related to cytokine signaling and secreted by the generated astrocytes under basal conditions, we analyzed the astrocyte-conditioned medium (ACM) collected from D95 astrocyte cultures using commercial cytokine antibody array. In concert with the transcriptomic data, only two compounds were detected at high levels (Figure 5d,e); monocyte chemoattractant protein-1 (MCP-1/CCL2) and tissue inhibitor of metalloproteinases (TIMP)-2 protein [46,47] were found in substantial amounts in the ACM. In the same way, a high level of MCP-1 was reported in both midbrain and spinal cord GFAP-expressing astrocytes generated from hPSCs [48]. TIMP-1 expression is stimulated by microglia and its level was low in D95 astrocyte cultures. The array was designed for 80 targets including, among other cytokines, the interleukin (IL)-6 family members IL-6 and Leucemia Inhibitory Factor (LIF) that promotes pluripotency in co-operation with CCL [49]. Only IL-8 and Insulin Growth-Factor-Binding Protein 2 (IGFBP2) were identifiable at a low level similar to TIMP-1, while the levels of others such as LIF were almost undetectable (Figure 5d). Respectively, mRNA expression of *CCL2* was significantly increased and *TIMP-2* showed a trend to increase in D95 astrocytes when compared to the expression of undifferentiated D0 neurosphere cells (Figure 5e). Consistent with our results, MCP-1 and TIMP-2 have been shown to be constitutively expressed in astrocytes [46,47].

## 4. Discussion

Astrocytes have become increasingly appreciated for their versatile role in maintaining proper homeostasis in the brain; hence, the number of methods for generating human astrocytes has increased. However, the complexity of astrocytes challenges the methodology for their production in vitro. The method described here provides added value to the methods published previously by a novel approach to facilitate differentiation of a homogenous astrocyte population for disease modeling application. Astrocyte differentiation followed the developmental generation of astrocytes, including neuroepithelial induction, regional patterning using morphogens, expansion of the progenitors, and astrocyte specification/maturation. The genetic profile of the differentiated astrocytes supported observations that regional identity of the astrocytes was acquired at the early stage of differentiation and persisted under in vitro conditions [50].

There are several issues, which complicate the optimization of astrocyte differentiation methods. First, the limits of astrocyte-specific markers make it difficult to characterize different subtypes of astrocytes. The most commonly used astrocyte marker, GFAP, is known to cover only about 15% of total astrocyte volume. Specifically, the fine processes and the interfaces between astrocytes do not in general express GFAP [51]. In addition, GFAP is not present in all astrocyte subtypes and its expression is regulated by the cells’ reactive state [52].

Secondly, the final stages of astrocyte maturation are considered to be mediated by neuronal activity that also promotes obtaining region-specific properties [15,17,53]. However, there is also evidence that astrocytes exhibit intrinsic mechanisms that determine their differentiation into distinct subtypes [54]. In any case, the cross-talk between neuronal and astrocyte populations is crucial for the formation of connections between astrocytes and neurons [55]. Glutamatergic signaling by neurons regulates both the morphology and expression of glutamate transporters in astrocytes [44,45,55]. Similarly, localization of glutamate transporter is induced by extracellular glutamate [56]. Neuronal input is required for the expression of EAAT2/GLT-1, whereas EAAT1/GLAST expression is increased as a result of neuronal activity [45]. This is consistent with our data and the low levels of EAAT2 observed in generated astrocytes indicated absence of neurons in the cultures.

The increase in glutamate transporters is mediated by soluble factors, and glutamate and cyclic adenosine monophosphate (cAMP) are implicated in the regulation of EAAT2/GLT-1 and EAAT1/GLAST. Expression of these glutamate transporters varies depending on the developmental stage as seen in morphologically distinct populations representing immature and mature cells [57]. With our method, astrocytes changed their gene expression profile and morphology upon culturing. However, D95 astrocytes retain many properties of immature cells, suggesting that, without neurons, astrocytes are in a more immature quiescent stage. However, D95 astrocytes were responsive to external stimuli, such as ATP. Considering the high level of purity, astrocytes generated with this method may be ideal for co-culture assays.

Key roles of astrocytes include production of extracellular matrix molecules and modulation of immune response and chemokine activity. We found that human D95 astrocytes constitutively expressed MCP-1 and TIMP-2, which were also secreted in higher amounts than other chemokines and cytokines under basal conditions. MCP-1 is a potent chemoattractant for monocytes/microglia and may promote neurotoxicity in damaged brain tissue. Its expression is increased in multiple sclerosis [58] and its involvement in the recruitment of inflammatory infiltrate into the CNS may be critical in a number of neurological disorders.

TIMPs are important for the maintenance of the proteolytic balance in the brain and their astrocytic expression regulates tissue break-down by neutralizing the effects of matrix metalloproteases (MMPs). In addition, TIMPs have MMP-independent functions and TIMP-2 has been connected to enhanced neuronal differentiation, while TIMP-1 is capable of influencing neuronal morphology. In our study, we found that differentiated D95 astrocytes expressed and secreted TIMP-2 under basal conditions in agreement with previously observed constitutive astrocytic expression of TIM-2. While TIMP-1 expression is induced by microglia, low levels of TIMP-1 in D95 astrocytes supported the high purity of our astrocyte population [46].

## 5. Conclusions

The method described here generates a human astrocyte culture of high purity, which makes the cells ideal for disease modeling to study the role of astrocytes under pathological conditions. Characterization of astrocyte cultures lays the foundation for the identification of disorder-related alterations. Gene expression studies supported forebrain identity of D95 astrocytes. Astrocytes expressed several developmental genes and showed transcriptional similarity with embryonic brain astrocytes, reflecting the developmental stage of astrocytes. The final astrocytic maturation with regional specificity likely requires interaction with neurons. Therefore, the present culture system might be particularly well suited for the research of neurodevelopmental disorders both as pure astrocyte as well as in co-culture assays. Astrocytes contribute to impaired neuronal function in many neurodevelopmental disorders [59,60,61] and patient-derived astrocyte models can make a significant contribution to studies of human brain function.

## Figures and Tables

**Figure 1 brainsci-11-00209-f001:**
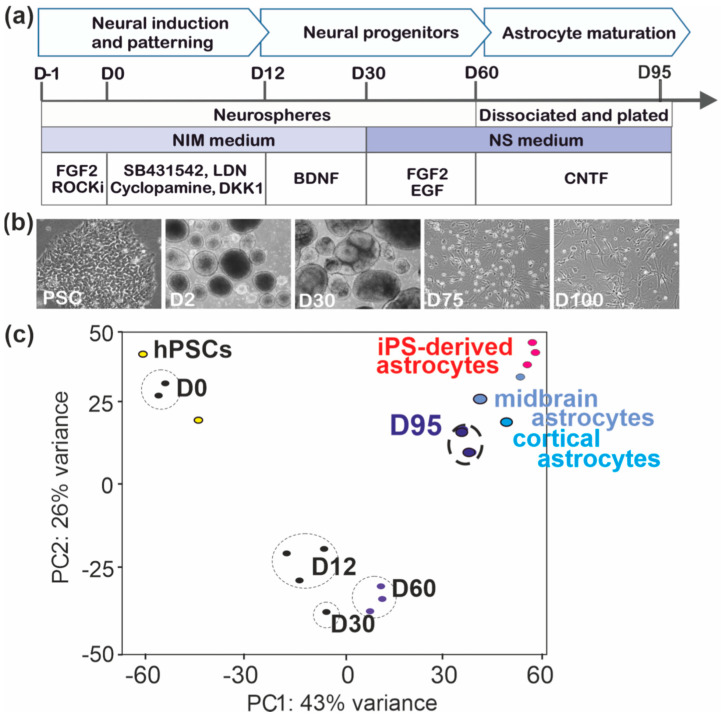
Differentiation of human pluripotent stem cell (hPSC)-derived astrocytes. (**a**) Schematic illustration of the astrocyte differentiation protocol. (**b**) Representative bright field images of cells at the different stages of differentiation; pluripotent cells (hPSCs), at Day 2 (D2), D30, D7), and D100 of differentiation. (**c**) Principal component analysis (PCA) of the RNA-seq data of hPSCs (*n* = 2) and cells at D0 (*n* = 2), D12 (*n* = 3), D30 (*n* = 2), D60 (*n* = 3), and D95 (*n* = 2) showed separated clustering of D95 astrocytes from the clusters of undifferentiated cells and D12, D30, and D60 cells. D95 astrocytes clustered together with human embryonic and hPSC-derived astrocytes published previously. Abbreviations: FGF2, fibroblast growth factor 2; ROCKi, ROCK inhibitor; DKK1, Dickkopf 1; BDNF, brain-derived neurotrophic factor; EGF, epidermal growth factor; hPSCs, human pluripotent stem cells.

**Figure 2 brainsci-11-00209-f002:**
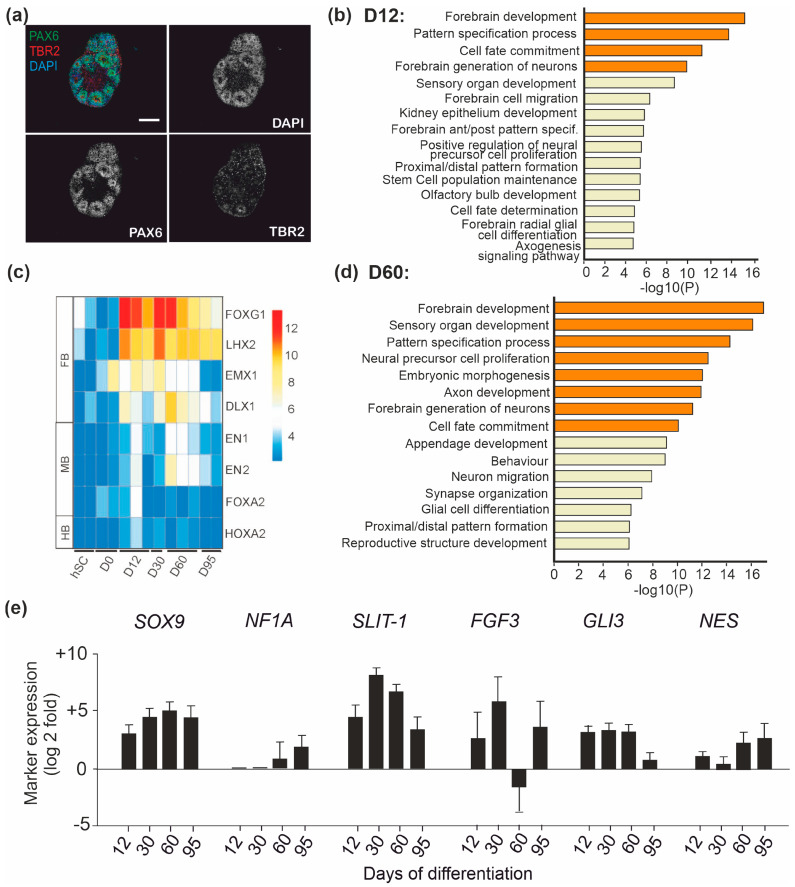
Patterning and fate determination of progenitors. (**a**) Progenitor differentiation was induced to neural lineages with dual SMAD inhibition and patterned towards forebrain identity. Representative images showing immunocytochemistry for PAX6 and TBR2 in neurospheres at D30 in vitro. (**b**) Ontology clusters of enriched transcripts (log2-fold, *p* < 0.05, *p* adjust < 0.1) in RNA-seq data at D12. (**c**) Heatmap showing expression of representative forebrain (FB), midbrain (MB), and hindbrain (HB) markers in RNA-seq analysis of undifferentiated hPSCs and during astrocyte differentiation at D12, D30, D60, and D95. (**d**) Ontology clusters of enriched transcripts (log2-fold, *p* < 0.05, *p* adjust < 0.1) in RNA-seq data at D60. (**e**) Expression of transcript encoding factors implicated in astrocyte fate determination and/or differentiation (*SOX9*, *NF1A*, *SLIT-1*, *FGF3*, *GLI3*, and *NES*) at D12, D30, D60, and D95 relative to the expression at D0 levels during astrocyte differentiation. Error bars represent s.d.

**Figure 3 brainsci-11-00209-f003:**
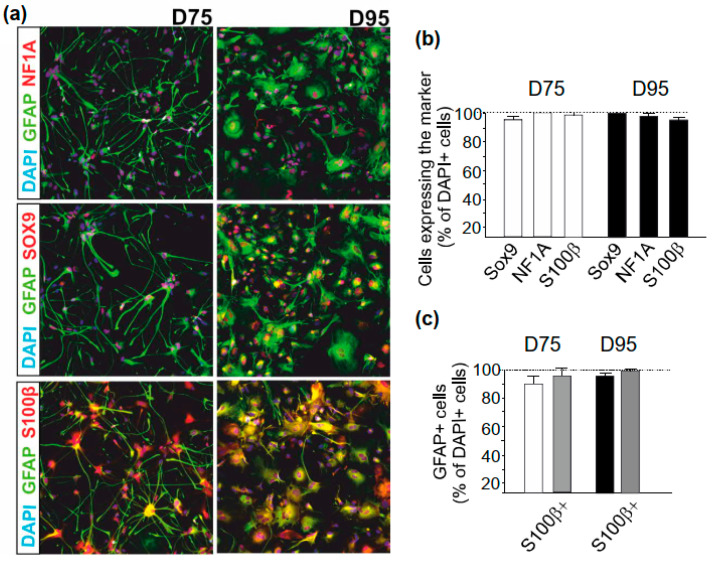
Immunostaining for astrocyte markers. (**a**) Representative images of astrocytes showing immunocytochemistry for astrocyte markers (overlay and separately for NF1A, SOX9, S100b, glial fibrillary acidic protein (GFAP), and DAPI nuclear staining) at D75 and D95. (**b**) Bar graphs showing percentage of Sox9, NF1A, and S100b immunopositive cells at D75 and D95. *n* = 3 cell lines, analysis of triplicate samples of each cell line. (**c**) Percentage of GFAP immunopositive cells at D75 and D95. *n* = 3 cell lines. Triplicate samples of each cell lines were analyzed. Data are expressed as mean ± SEM.

**Figure 4 brainsci-11-00209-f004:**
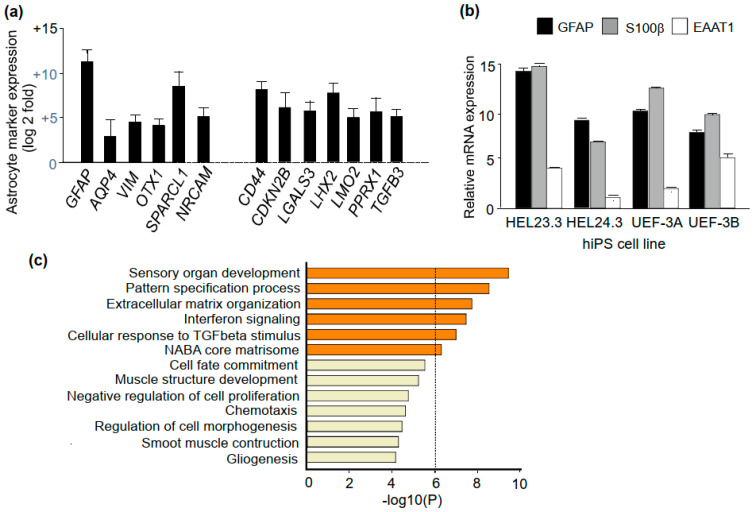
Molecular signature of D95 astrocytes. (**a**) Enriched expression of astrocyte markers relative to neurosphere expression in RNA-seq data of D95 astrocytes (log2-fold, *p* < 0.05, *p* adjust < 0.1; duplicate samples). (**b**) Expression of GFAP, S100b, and EAAT1 mRNA in D95 astrocytes. *n* = 3 cell lines. (**c**) Enriched ontology clusters in D95 astrocytes.

**Figure 5 brainsci-11-00209-f005:**
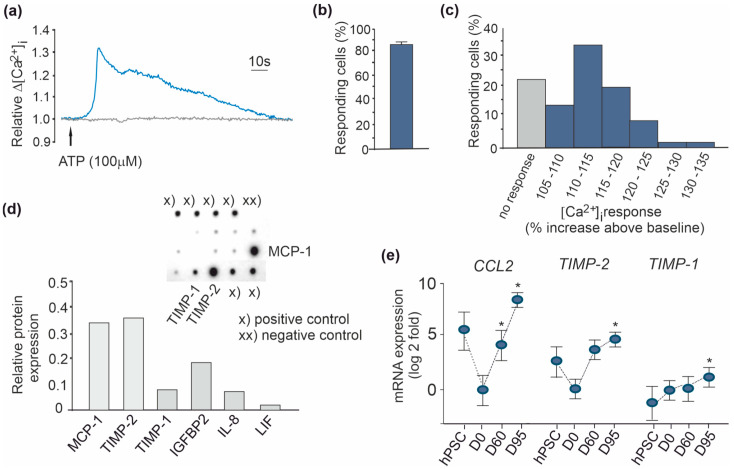
Functional testing of D95 astrocytes. (**a**) Representative single cell response to ATP in intracellular calcium recordings of astrocytes, (**b**) the number of cells responding to ATP (data are expressed as mean ± SEM), and (**c**) the distribution of cell responses suggesting high purity of the astrocyte population. *n* = 5 cell lines used in repeated studies, responses of 75–100 cells were measured in each experiment. (**d**) Monocyte chemoattractant protein-1 (MCP-1), tissue inhibitor of metalloproteinases-2 (TIMP-2), tissue inhibitor of metalloproteinases-1 (TIMP-1), Insulin Growth-Factor-Binding Protein 2 (IGFBP2), interleukin-8 (IL-8), and Leucemia Inhibitory Factor (LIF) amounts in astrocyte-conditioned medium (ACM). (**e**) Expression of *CCL2/MCP-1*, *TIMP-2*, and *TIMP-1* relative to neurosphere expression (D0) in RNA-seq data of aged D60 and D95 cells and undifferentiated hPSCs. * *p* < 0.05, *p* adjust < 0.1.

## Data Availability

The data presented in this study will be openly available in GEO.

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
