# Peer review of "Generation of the Human Pluripotent Stem-Cell-Derived Astrocyte Model with Forebrain Identity"

_brainsci, 2021, doi:10.3390/brainsci11020209_

Round 1
Reviewer 1 Report
Comments to Authors:
In the present report, Dr. Peteri et al. describe the development of a detailed and rigorous method for efficiently and reproducibly generating high purity astrocytes with forebrain identity. These studies represent an important advance in the field, as prior methods for generating astrocytes, while simpler and quicker, result in populations containing large numbers of neural cells. In addition, the ability to obtain astrocytes with properties unique to those present in the forebrain could prove very useful for modeling of diseases in this brain region that have glial involvement. The studies are well designed and executed, the data are sound, and the manuscript is well written. However, the authors need to provide a bit more detail on what properties of the derived astrocytes indicate that they are of forebrain identity and why this particular subset/subpopulation of astrocytes is particularly valuable for disease modeling, i.e., what diseases could be modeled with these astrocytes. This additional information would greatly strengthen the manuscript and help set it apart from prior work in this field.
Author Response
Response to the Reviewer Comments
Reviewer #1.
We thank the Reviewer for thoughtful assessments of the manuscript. The manuscript has been revised to address concerns as follow:
Response. We have revised the text in the Conclusion (lines 382-393) to summarize that the gene expression changes support forebrain identity of astrocytes at the immature stage likely due to the absence of neurons and that the astrocytes are therefore particularly well suited for research of neurodevelopmental disorders. We added three new references describing astrocytic abnormalities in neurodevelopmental disorders.
Reviewer 2 Report
In the current manuscript, Peteri et al. demonstrates that culture method to generate high purity astrocytes from hPSCs. The method was reproducible across several hPSC lines, and fully differentiated astrocytes show intracellular calcium responses and constitutively express some chemokines demonstrating its functionality. Overall, the findings are informative for researchers in the field, however some corrections will improve this manuscript.
Comments
1. Typo p2 l94, 2.2. Immunocytochemistry.
2. Add the method of cytokine antibody array for ACM. Which array do authors use?
3. Change color to distinguish iPS-derived astrocytes and midbrain cortical astrocytes in Figure 1c. Also add each sample numbers in figure legend since some points are hard to identify. i.e. hPSCs (n=2), D0 (n=2).
4. Figre2c and 2d are opposite, and D30 should be D60 in Figure 2e.
5. Figure 2b. Relative RNA expression to what? Difficult to understand in glance. Can be omit since staining data in Figure 2a is more informative.
6. Figure 2f, figure legend. SLIT-1, FGF3, NES are not ‘transcription factors’.
7. Figure 5d must be improved, since this data is the most novel and most important information to be demonstrated. Besides MCP-1, TIM-2, TIMP-1, and LIF, what kinds of other cytokines are listed in the array? What about the amounts for other IL-6 family genes? And what is the basal expression level of MCP-1, TIM-2 in hPSC?
8. Figure 5e. Add the basal expression level (D0) in the graph.
Author Response
Reviewer #2.
We thank the Reviewer for careful assessment of the manuscript. The manuscript has been revised to address each concern as follow:
- Typo p2 l94, 2.2. Immunocytochemistry.
Response to point #1. The typo is corrected.
- Add the method of cytokine antibody array for ACM. Which array do authors use?
Response to point #2. Human Cytokine Antibody Array (Abcam, Ab133998) was used to screen cytokine levels in ACM. We have added a new paragraph (Materials and Methods; lines 153-164) “2.6. Cytokine array” that describes collection of ACM and cytokine detection using the Abcam Array.
- Change color to distinguish iPS-derived astrocytes and midbrain cortical astrocytes in Figure 1c. Also add each sample numbers in figure legend since some points are hard to identify. i.e. hPSCs (n=2), D0 (n=2).
Response to point #3. In the revised Figure 1c, iPS-derived astrocytes are shown in red and midbrain/cortical astrocytes shown in blue to improve resolution. In addition, the sample numbers are shown in the figure legend.
- Figre2c and 2d are opposite, and D30 should be D60 in Figure 2e.
Response to point #4. The panels have been reorganized and D30 has been corrected to D60 corresponding to the text body in the revised Figure 2.
- Figure 2b. Relative RNA expression to what? Difficult to understand in glance. Can be omit since staining data in Figure 2a is more informative.
Response to point #5. We have omitted Figure 2b.
- Figure 2f, figure legend. SLIT-1, FGF3, NES are not ‘transcription factors’.
Response to point #6. In the revised manuscript “pro-glial cell transcription factors” is replaced by “factors implicated in astrocyte fate determination and/or differentiation (SOX9, NF1A, SLIT-1, FGF3, GLI3, and NES)”
- Figure 5d must be improved, since this data is the most novel and most important information to be demonstrated. Besides MCP-1, TIM-2, TIMP-1, and LIF, what kinds of other cytokines are listed in the array? What about the amounts for other IL-6 family genes? And what is the basal expression level of MCP-1, TIM-2 in hPSC?
Response to point #7. The array was designed for 80 targets. Among other cytokines there were the interleukin (IL)-6 family members IL-6 and Leucemia Inhibitory Factor (LIF). Only IL-8 and Insulin Growth Factor Binding Protein 2 (IGFBP2) were identifiable at a low level similar to TIMP-1, while the levels of others such as LIF were clearly lower. The text is revised as follows: “The he array was designed for 80 targets, including among other cytokines the interleukin (IL)-6 family members IL-6 and Leucemia Inhibitory Factor (LIF) that promotes pluripotency in cooperation with CCL [50]. Only IL-8 and Insulin Growth Factor Binding Protein 2 (IGFBP2) were identifiable at a low level similar to TIMP-1, while the levels of others such as LIF were clearly lower (Fig. 5d).” In addition, the IL-8 and IGFBP2 levels are shown in the revised Figure 5e.
- Figure 5e. Add the basal expression level (D0) in the graph.
Response to point #8. Basal expression levels (0-level) are added to the graph of Figure 5e. In addition, the expression levels in hPSCs relative to D0 are shown in the revised Figure 5e.